# A Model for Examining Challenges and Opportunities in Use of Cloud Computing for Health Information Systems

Ahmad Al-Marsy , Pankaj Chaudhary * and James Allen Rodger

Information Systems and Decision Sciences Department, Indiana University of Pennsylvania, 1011 South Drive, Indiana, PA 15705, USA; dnkw@iup.edu (A.A.-M.); jrodger@iup.edu (J.A.R.)
* Correspondence: pankaj@iup.edu

**Abstract:** Health Information Systems (HIS) are becoming crucial for health providers, not only for keeping Electronic Health Records (EHR) but also because of the features they provide that can be lifesaving, thanks to the advances in Information Technology (IT). These advancements have led to increasing demands for additional features to these systems to improve their intelligence, reliability, and availability. All these features may be provisioned through the use of cloud computing in HIS. This study arrives at three dimensions pertinent to adoption of cloud computing in HIS through extensive interviews with experts, professional expertise and knowledge of one of the authors working in this area, and review of academic and practitioner literature. These dimensions are financial performance and cost; IT operational excellence and DevOps; and security, governance, and compliance. Challenges and drivers in each of these dimensions are detailed and operationalized to arrive at a model for HIS adoption. This proposed model detailed in this study can be employed by executive management of health organizations, especially senior clinical management positions like Chief Technology Officers (CTOs), Chief Information Officers (CIOs), and IT managers to make an informed decision on adoption of cloud computing for HIS. Use of cloud computing to support operational and financial excellence of healthcare organizations has already made some headway in the industry, and its use in HIS would be a natural next step. However, due to the mission's critical nature and sensitivity of information stored in HIS, the move may need to be evaluated in a holistic fashion that can be aided by the proposed dimensions and the model. The study also identifies some issues and directions for future research for cloud computing adoption in the context of HIS.

**Keywords:** health information system; electronic health records; DevOps; cloud computing; continuous integration; continuous deployment



## 1. Introduction

Information systems (IS) are where Information technologies and Business meet. An information system can be defined as the group of interrelated components that collect, process, store, and distribute information [1]. A Health Information System (HIS) is an IS for the health care sector. Such systems provide benefits from Health Information Technologies (HIT) and components, to produce Electronic Health Records (EHR) as the primary information product. HIS is a core business system for any health organization in addition to other IS providing financial and other operational functionalities. An HIS can enhance healthcare organizations' operations and performance significantly by reducing cost and improving outcomes [2]. Due to their many benefits, HIS are considered crucial to healthcare organizations globally [3].

The information stored in HIS is also used to support decision making and enhancing organizational controls. Business intelligence (BI) and Clinical Decision Support Systems (Clinical DSS) benefit from information stored in HIS to provide both business and medical decisions to improve organizational operations and performance. Patient data in healthcare has the characteristics of 3Vs of big data, volume, variety, and velocity. Artificial intelligence

(AI) and Machine Learning (ML) models can be applied to this patient data to enhance decision-making and provide faster ways for diagnostics and discerning patterns. The insights gained from this analysis can significantly enhance the quality of healthcare [4–7].

Given the critical role of IS, many methodologies and paradigms have been suggested to improve agility, efficiency, and availability of IS in general. One of these popular paradigms is cloud computing. IS agility in the current environment is critical for the IS to keep up with the ever-changing business requirements. Technological and environmental changes (e.g., regulatory environment, pandemics) require fast changes in IS since most critical business operations rely on IS. HIS face the same challenges of IS agility as well [8]. Cloud computing helps in IS agility in addition to benefits of reducing costs and providing on-demand, bleeding edge, high-performance, and highly available infrastructure, platforms, and software. Cloud computing provides tools and feature sets to achieve Continuous Integration (CI) and Continuous Delivery (CD) to facilitate the implementation of DevOps within an organization. DevOps is a proven methodology to enhance the agility of HIS and IS in general [9,10] by providing the capability to roll out new and enhanced features in a short time duration in an incremental fashion. With these technologies and concepts, HIS may be easier to maintain and upgrade. These also give new HIS competitive advantage in being faster to the market with new features and functionality. HIS need to be improved so that they are more scalable, highly available, and fault tolerant. All these benefits can again come from the use of cloud computing. However, using cloud computing has its own challenges. These include cost, operations, security, and a complex and fairly new business model amongst others.

This study reviews both academic and significant practitioner literature in the field of cloud computing and HIS. It also draws upon the expertise of one of the authors with 6+ years of technology and consulting experience in this area and interviews with experts in Healthcare informatics or Cloud Computing. Based on the review, author expertise, and interviews, three primary dimensions affecting HIS implementation in the cloud are identified. These dimensions are (1) financial performance and cost; (2) IT operational excellence and DevOps; and (3) security, governance, and compliance. Each of these dimensions consist of certain drivers as well as major challenges regarding adoption of cloud-based HIS. The research provides a model that can be used by practitioners to aid decision-making and can be empirically verified. It also forwards some additional research issues that can be examined by researchers, suggestions, and direction for future work along with some important recommendations.

*Problem Statement and Significance*

Identification of factors that affect the adoption of cloud computing technologies in implementation of HIS has received insufficient attention in the academic literature, and hence there is paucity of studies with systematic analysis and rigor. This study aims to fulfill those gaps through its proposed model of study. It proposes a framework for study of the primary drivers and challenges in the aforementioned dimensions and presents a model. This model proposed has face validity, content validity, and nomological validity and may be confirmed through an empirical analysis. The model and subsequent analysis may provide a better understanding of the challenges and possible solutions in the adoption of cloud computing in the healthcare sector and more specifically in relation to HIS.

The results of this study are targeted towards executive management of health organizations, especially senior clinical management positions like CTOs, CIOs, and IT managers so that they can make an informed decision about cloud computing adoption. The study elaborates on how HIS functionality can be enhanced through the of cloud computing. Improvements in the operational excellence and financial performance of healthcare organizations using cloud computing technologies that overcome the limitations of in-place legacy information systems are also examined. This study will serve as an aid for future researchers and adopters through its detailed examination of drivers and challenges that

are pertinent to the adoption of cloud computing technologies for HIS and will also help in managerial decision making.

The paper is structured as follows. Section 2 provides an overview of the HIS. Section 3 provides a background of cloud computing and its benefits. Section 4 provides a detailed treatment of the dimensions of financial performance and cost; IT operational excellence and DevOps; and security, governance, and compliance, and associated drivers and challenges pertinent to each of these dimensions. Based on the discussions, a set of hypotheses are proposed with respect to each driver and challenge and its effect on the adoption of cloud computing in HIS. Section 5 presents a confirmatory structural equation model along with single item measures for each of the challenges and drivers and readiness of adoption of cloud computing for HIS. It also outlines some gaps in cloud computing research that are also pertinent to cloud-based HIS, which need further attention and may be taken up by other researchers or the industry. Section 6 gives details on the proposed use of the model for an academic study in Palestine and India. Finally, Section 7 provides conclusions and possible avenues for future research.

## 2. Overview of Health Information Systems

According to Yusof et al. [11] HIS can be defined as an IS that is used in a Healthcare setting and consists of multiple components that are related and dependent on each other. The main functions of an HIS are to gather, process, store, and distribute information of healthcare providers and their stakeholders, thereby improving the efficiency and effectiveness of healthcare [11,12]. HIS consists of humans, procedures, technology, and the interaction between them [13]. HIS may also be described as a socio-technical subsystem in hospitals that consist of information processing systems and human or technical staff that perform the processing, supporting hospital affairs [14].

The 1960s saw the introduction of computerized systems targeting healthcare [15]. The aim of these systems was to automate some operational and administrative tasks and manage staff and patient financials. The evolution in medicine and medicaid sciences and the introduction of evidence-based medicine led to increased focus on medical records [15,16]. The advances in IT and systems in 1970s led to the introduction of EHR. Nguyen el al. [17] define EHR as the virtual record of all health-related events of an individual during his whole lifespan that includes but is not limited to medication history and allergies, admissions, and/or outpatient clinic visit. International Organization for Standardization (ISO) defines EHR as a repository of information regarding the health status of a subject of care, in computer processable form [18]. Subsequently, decision support features were included in HIS to provide better diagnostics and treatment [15,19–22]. HIS has also benefited from the wider adoption of IT and the improvements in the communications and computer networks. The availability of internet through multiple data networks has led to the introduction of many centralized and decentralized systems that facilitate information sharing and distribution between different parties. These systems are termed as Health Information Exchange (HIE). An HIE is a regional or national organization that facilitates the electronic sharing of healthcare data among providers, facilities, health information organizations, government agencies, and patients [23]. HIE allow healthcare organizations to appropriately access and securely share a patient's medical information electronically, before, during, and at the conclusion of an episode of healthcare [14,23]. HIS play an important role in this sharing through the sharing of EHR through HIE. However, there are several interoperability challenges when it comes to sharing EHR that still need to be resolved using standardization at several layers of technology, data, human, and institutions [24]. Even when interoperability may not be an issue, HIEs report that EHS vendors and health systems engage in information blocking [25]. Cloud-based HIS may alleviate some of the technical concerns of sharing EHR data via HIEs, however, the idea of collecting personal health information into a single repository elicits serious security and privacy concerns from patients, and control and ultimate usage concerns from providers [26]. Currently, with the expansion of cloud services and providers, concepts like

e-health and Personal Electronic Health Records (pEHR) have been introduced to increase patient empowerment [16,20,27,28].

Due to the advancement in IT, and medical and medicaid sciences, current HIS have become more complex and capable [12] and are expected to perform the following functions [14,15]:

- Clinical information management: This relates to patient information and includes EHR, information extracted from the internal submodules, or the integration with other third-party systems. These sources can be from laboratories, radiology, inpatient, outpatient, emergency department, or pharmacies. Clinical decision support, knowledgebase, and order communication systems are also considered in this category.
- Operational management: This includes all non-clinical-related functions such as financials—billing and payroll; and human resources, which includes rosters and shift management. Operational management also includes procurement, supply chain management, maintenance, and engineering.
- Strategic decision support: This helps higher management in strategic planning and better organizational control. This may include better marketing insights, market trends, and competitor behavior.
- Electronic networking [2,16]: This includes all interfaces that involve health information movement over network. Examples are web-based telemedicine and e-health and pEHR that helps patients to have more control and empowerment over their health profiles remotely, away from the healthcare organizations' physical location. Also, network integration with other systems and their parties can be considered in this category.

According to Gaardboe et al. [5], healthcare organizations having IS have shown better profit margins than those who do not, and this is due to better operational performance. For example, Kros et al. [29] and Kwon et al. [30] discussed the importance of IS in supply chain management in healthcare. The study done by Kwon et al. [30] found that IS-enabled supply chain management has a positive effect on efficiency by reducing per-patient cost. The study also argues that supply chain management systems in healthcare enhance patient treatment, by focusing on supplier relationship management, logistics operational tools, and process improvement.

## 3. Cloud Computing

New technologies like virtualization, grid computing, and software-defined datacenters, which includes storage and networking, have become mainstream and have resulted in a paradigm shift in the IT industry. Virtualization enables computer hardware to run different operating systems and application on the same hardware by abstracting the existing hardware from the application and emulating the needed hardware [8]. Grid computing is the distribution of processing between multiple computer systems to achieve enough processing power to perform the needed task [28]. These technologies and others have led to the concept of cloud computing [1,15,31,32]. Cloud computing is a service model that involves on-demand access to resources provided by a remote vendor usually over the Internet [33], though for performance reasons some organizations may lease a connection to the nearest provider facility. The resources provided by cloud service providers are usually shared pools or virtualized resources [1]. These resources can be categorized into three types, based on their nature. Infrastructure as a Service (IaaS), where the resources provided are infrastructure resources; Software as a Service (SaaS) in which software is the provided resource; and Platform as a Service (PaaS) that includes a total platform as the hosted resource [28,33].

Cloud computing can have a great impact on improving the healthcare services as well as the innovative effect of how it affects the other sectors [34]. Marston et al. [8] suggests that cloud computing is a result of two key trends in IT.

First is that cloud computing provides efficiency. Cloud providers usually offer highly available and scalable solutions. Healthcare organizations can benefit from these features

by reducing the capital cost associated with an on-premises implementation of a datacenter [35]. Datacenters for healthcare organizations are expensive because of the lifesaving nature of these organizations and the need for the datacenter to support this mission. To accomplish this, assurance high availability and zero downtime of services is needed. High availability and zero down time are complex technical problems requiring many resources, and hence an expensive proposition. In a non-cloud scenario, organizations tend to have multiple levels of redundancy and high availability and disaster recovery. This may involve multiple datacenters or co-located equipment in datacenters run by others to achieve the service availability and compliance with governmental regulations and national or international standards [34,36,37]. This requires substantial capital expenditure. Newly established organizations can benefit from the capital savings by moving to cloud and direct the savings to investment in core medical business, like medical personnel. Established organizations can migrate gradually into cloud as their current equipment is fully depreciated. They can use the hybrid cloud design to connect their current datacenters to the cloud for expansion and disaster recovery [38]. Usage of cloud computing lowers the operational costs by eliminating the costs needed for datacenter operations like electricity and cooling too. It also reduces the costs of technical and IT personnel needed to manage and operate the infrastructure [8]. The on-demand nature of cloud computing services is another example of how cloud computing provides IT efficiency [31,32]. Cloud services allow organizations to acquire their infrastructure and services as needed—on demand. As the demand for the system increases, additional infrastructure resources are provisioned and deployed automatically, and they are terminated automatically once the demand goes down again. This dynamic resource allocation reduces the operational and fixed cost of the IT infrastructure for healthcare organizations.

Second is that Cloud computing as suggested earlier also provides IS and business agility. Agility in execution is the hallmark of cloud services. Cloud services provide organizations with features and functions to improve business operations. High availability and disaster recovery features of cloud services are critical for healthcare [34] and allow for quick recovery. Service providers also may offer prebuilt and production ready tools and platforms that increase application development speed of the organization. These can be integrated with existing systems through well-documented Application Programming Interface (API) and provider support. For example, tools like Tableau [6] offer Business Intelligence (BI) platform over cloud. In case an organization is in need of high processing power for AI, ML, and Business Intelligence (BI) purposes that exceeds the capacity of the organizational infrastructure, cloud services provide parallel computing and big data solutions [39].

There are many examples for the usage of cloud computing in the healthcare environment in addition to HIS. Health care organizations can use Microsoft Office 365 and Google GSuite (Google, Mountain View, CA, USA). Microsoft and Google also provide several PaaS and SaaS solutions to their customers. Amazon Web Services (Amazon, Seattle, WA, USA) Microsoft (Redmond, WA, USA) and others provide IaaS for software-defined data centers. Koutsouris et al. [2] discussed the use of cloud computing for pEHR. Mobile cloud computing-based stroke healthcare system [40] is an example of a cloud computing-based system that is used to identify stroke patients with cardioembolic and cryptogenic subtypes using mobile apps. There are several other examples of implementation of cloud-based healthcare systems in different countries in Americas, Europe, and Asia [16,28,35].

## 4. Drivers and Challenges for Cloud Based HIS

The dimensions, drivers and challenges they incorporate have been arrived at based on the extensive review of the academic and practitioner literature, extensive experience of one of the authors in this area, and interviews with experts in healthcare informatics or cloud computing fields. These experts were chosen based on the number of years of experience in their fields, and if they are holding some of the well-respected certificates in the cloud computing or health informatics areas. The interviews also included some clinical,

operational, or financial top management of some healthcare organizations. Interactions with attendees and information in different sessions at AWS reinvent, Kubecon, and Cloud Native conferences were also used in formulating the proposed model. Insights from the interviewees have also been incorporated into Sections 2–5 and 7. Table 1 provides a profile of the interviewees.

**Table 1.** Profile of the interviewees.

| Company | Nature of Responsibility of the Interviewee | Industry | Product/Services of the Company |
|---|---|---|---|
| 1 | Chief Internal Auditor | Investment | Palestinian investment firm that invested in many local hospitals. |
| 2 | Consultant | Financial and security Auditing | One of the Big four accounting firms that participated in external auditing of private hospitals. |
| 3 | Chief Product Officer (CPO) | SaaS, Hospitality | Hospitality SaaS Cloud hosted provider |
| 4 | Chief Nursing Officer (CNO) | Healthcare | Privately held hospital with locally hosted HIS |
| 5 | CEO | Healthcare | Privately held hospital with locally hosted HIS |
| 6 | Advisory and board member | SaaS, Healthcare | Hybrid cloud SaaS provider for HIS complementary functions |
| 7 | Director of design and architecture | SaaS, Hospitality | Hospitality SaaS Cloud hosted provider |
| 8 | VP of Engineering | SaaS, Healthcare | Hybrid cloud SaaS provider for HIS complementary functions |
| 9 | CTO | IT solutions | Outsourcing company with multiple, on-premises and cloud hosted solutions. |
| 10 | CTO | Security | IT Security firm with clients in healthcare field |
| 11 | Chief Financial Officer (CFO) | Healthcare | Privately held hospital with hybrid cloud HIS |
| 12 | IT Manager | Healthcare | Public hospital with on-premises HIS |
| 13 | Technology Evangelist | IT solutions | A well-known open-source IT solutions company helping many clients in many sectors to adapt cloud computing. |
| 14 | QA Manager | Healthcare | Public hospital with on-premises HIS |
| 15 | CIO | Healthcare | Privately held regional system in US transitioning to Cerner hosted HIS |
| 16 | Chief Medical Officer (CMO) | Healthcare | Large specialty hospital in India without an EHR and separate laboratory systems |

This section provides a detailed analysis of opportunities and challenges for cloud implementation of HIS. The dimensions of financial performance and cost; IT Operational excellence and DevOps; and security, governance, and compliance are examined along with their drivers and challenges. Figure 1 provides details for these dimensions.

## Financial performance and cost

**Driving factors:**

- Total Cost
- Cloud pricing models

**Challenges:**

- Unexpected or hidden cost components
- Unmonitored on demand usage
- Financial effects of service outage

## IT Operational excellence and DevOps

**Driving factors:**

- Offloading IT operations and focus on business.
- Automatic scaling
- Ready to use SaaS and PaaS
- DevOps and automation
- Availability of opensource alternatives for cloud management
- Containers and Microservices

**Challenges:**

- Vendor Lock
- Complexities in design and operation phases of systems
- Skillset and experience requirements

## Security, governance, and compliance

**Driving factors:**

- Inheritance of security controls and benefits of providers
- Obtain and maintain HIPAA and HITECH compliance with less cost

**Challenges:**

- Shared responsibility model
- Share security vulnerabilities with vendor

**Figure 1.** Cloud computing opportunities and challenges.

*4.1. Financial Performance and Cost*

Operational excellence and efficiency are some of the main advantages of using HIS, which leads to increased financial performance [41,42]. Financial performance of health organizations can be enhanced by using cloud computing to modernize HIS through efficiency gains and operational and capital cost reduction as discussed previously. However, the use of cloud may also have unexpected or hidden cost components such as migration costs and application management costs, which may increase costs dramatically. From the cost perspective, pay-as-go and on-demand pricing model [28,43] of the cloud reduces the startup cost for design, implementation, and management of the IT infrastructure. This pricing model has attracted many firms [8,28], especially startups to adopt cloud computing. The literature mostly focusses on how business has shifted to cloud computing for cost benefits but has less focus on how the pay as go model may lead to high-cost drifts especially if it is unmonitored [44]. Empirical studies are needed to compare costs between well architected on-premises datacenters with a proper expansion forecasting to host HIS as compared to the pay-as-you-go model of the cloud providers. Also, more studies on customer satisfaction with cloud providers' billing and costs in multiple industries including healthcare need to be conducted to measure the cost benefits of cloud solutions and to have a reliable estimation of how much saving cloud computing adoption can lead to.

Another area that is worth investigating is the various pricing schemes provided by the cloud providers. Pay as you go and on-demand pricing have been the primary focus in

the literature; empirical studies are needed to study the adoption models of other pricing schemes like reservation, leasing, and excess capacity usage. Reservation and leasing are a commitment to use predefined computer resources for a specific duration, and excess capacity usage is the use of idle and already running computer resources until they were reclaimed when needed by the providers [45,46]. Cloud providers market these schemes as they provide a respected amount of savings [45,46]. Organizations that have conducted a detailed load study of their resource requirements can benefit from such schemes through various automated provisioning tools.

A reduction in operational costs needed to manage and maintain internal datacenters [28,43] is another advantage of using cloud computing, however, more focus may be needed on the financial effects in situations such as service outage since the maintenance and support are provided by the cloud vendors and not by the institution itself. Cloud service outages are well known [47] and make big news as they happen. However, the big news is usually in the media when consumer end services like Gmail and others are affected. Statistics pertaining to corporate downtimes are not available since the data may not be disclosed. In the case of HIS, even a few minutes of downtime can create issues. When the HIS is hosted outside the organization, then the amount of uncertainty and feeling of loss of control can compound the situation. Information in HIS is lifesaving and availability of data should be guaranteed, therefore, more empirical studies need to be conducted on the satisfaction of the uptime, service availability, and support levels provided by the cloud vendors in the context of healthcare institutions.

Interviews with financial leaders and consultants (1,2,11) helped in arriving at the financial performance and cost factors. The CFO (11) focused on the yearly budget for cloud computing while the financial auditors (1) focused more on the amendments to budgets because of unexpected costs and the unmonitored usage of cloud computing. Session speakers in AWS Reinvent, Kubecon, and Cloud Native conferences along with the technology evangelist (13) shared their experiences with clients that incurred financial impact due the outages in the cloud services. This was due to bad implementation or uncontrolled outage from the service provider. This point was validated with the interviewees (4,5,15,16), with their explicit opinions and concerns on the matter.

From Figure 1, the following hypotheses are related to financial performance and cost in adoption of cloud computing for HIS:

**Hypothesis 1 (H1).** *Financial performance and cost influence cloud computing adoption for HIS.*

**Hypothesis 1-1 (H1-1).** *Total cost of cloud service will positively influence cloud computing adoption for HIS.*

**Hypothesis 1-2 (H1-2).** *Availability of different pricing models of cloud service will positively influence cloud computing adoption for HIS.*

**Hypothesis 1-3 (H1-3).** *Hidden costs of cloud service will negatively influence cloud computing adoption for HIS.*

**Hypothesis 1-4 (H1-4).** *Unexpected costs of on-demand pricing model will negatively influence cloud computing adoption for HIS.*

**Hypothesis 1-5 (H1-5).** *Financial effects of service outage will negatively influence cloud computing adoption for HIS.*

*4.2. IT Operational Excellence and DevOps*

Application performance can be easily enhanced using cloud computing. Cloud providers make the process of upgrading hardware and operating systems seamless without the need to purchase new equipment or wait for hardware depreciation [48]. Cloud

computing also allows for on-demand and automatic scaling up or down of resources based on the business use case to accommodate with the needed performance levels based on usage or periodic demand of the applications [49]. Usually, the information and benchmarks for the performance of IS in general are provided in a marketing context and have lacked a scientific basis. They are even more scarce in the context of health care and HIS. More empirical studies are needed to study the performance gains from the use of cloud infrastructure compared to traditional and on premises data center. Additionally, studies need to focus on performance gains and cost–benefit ratio since hardware upgrades are done by the providers. Lastly, auto scaling features and their impact on application performance need to be assessed in more detail and compared with gains from well-forecasted hardware upgrades, with more longitudinal studies to measure the financial impact of these solutions over the long term.

Cloud providers are always competing to attract customers by providing their own ready-to-use services and out-of-the-box SaaS and PaaS solutions that will help customers to deliver faster with few details on the underlying components. From a technical point of view, this may lead to a stronger cloud provider coupling, and thus vendor lock-in. As an alternative, many [8,10,50] have suggested the usage of cloud agnostic opensource software tools to facilitate the system movement and migration. Examples of these software tools include Apache Cloudstack (https://cloudstack.apache.org/ (accessed on 26 January 2021)), Openstack (https://www.openstack.org/ (accessed on 26 January 2021)), ManageIQ (https://www.manageiq.org/ (accessed on 26 January 2021)), Cloudify (https://cloudify.co/ (accessed on 26 January 2021)) etc. These tools have been effectively used to decrease vendor lock-in. In this area, comparison between opensource software and cloud providers can be conducted to compare the implications of cloud provider coupling to adoption of opensource solutions from cost, operations, and maintenance point of views, and to more precisely measure the operational overhead and costs of maintaining the in-house open-source cloud agnostic solutions.

DevOps [9,51] is an established paradigm in the IT industry for continuous delivery of application solutions to meet market and customer demands. DevOps culture and facilities related to cloud computing not only improve the performance of the application, but also increase the agility and resilience of IS and reduce the financial risks related to downtime and data losses [51]. DevOps deals with the goal of having the development and operations teams work closely together for achieving rapid and continuous release cycles [52]. Application delivery and quality have improved by the way DevOps methodologies develop, build, deploy, and deliver applications. Applications run in containers that are lightweight, isolated, and secured-system processes that share the same operating system to better utilize the system resources [53].

DevOps enhances and speeds up the software development lifecycle through: continuous integration (CI), which includes the planning and source code version control; continuous delivery (CD) of ready-to-release builds/deployment to production; continuous testing; observability through continuous logging; infrastructure monitoring and tracing; and application performance monitoring. Figure 2 [54] summarizes the integration between application development and operations, which is the core concept behind DevOps. Easy and fast provisioning of application resources is a key advantage of cloud computing coupled with DevOps [9,28]. The usage of both of Infrastructure as Code (IaC) [51] and containerization techniques [49,55] provide agility in provisioning and managing new applications and features, as well as infrastructure needed to host any IS. They also enable quick deployment and scaling of applications.

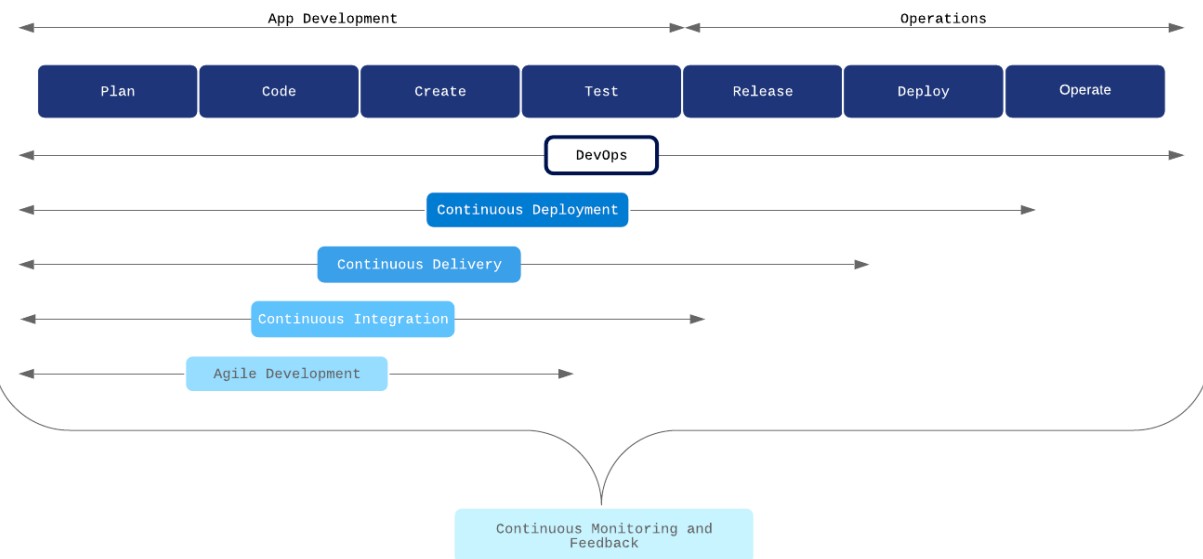

**Figure 2.** DevOps Methodology components.

By introduction of DevOps and containers, the continuous integration and continuous delivery of the system, efficiency, and success is more likely [9]. This is extremely beneficial for HIS. Since an HIS implements quite a number of features and functionality, it is ideal to divide those into smaller subsystems, which can be implemented using the concept of microservices [10]. Cloud HIS exist [35,40], and HIS implementation and design frameworks can be inferred from these implementations. However, a more focused approach for designing and implementing a framework for HIS using DevOps and microservices architecture is needed [56,57]. More empirical studies in the HIS context can be conducted to measure software lifecycle Key Performance Indicators (KPIs) of HIS implementations using DevOps. The KPIs may include delivery times, compliance with requirements, performance, availability, and scalability. KPIs of a microservices architected HIS implementation may be compared to the traditional implementations. Additionally, reimplementation and refactoring of already implemented traditional HIS systems would be a welcomed experiment to measure and identify success factors of traditional vs. microservices architecture. Lastly, complexities in design and operations phases of systems along with skillset and experience needed for successful implementation and operation of HIS need to be identified.

Complexities in designing cloud architectures especially in a situation like multicloud tenancy and hybrid cloud configurations can be daunting. That coupled with lack of skilled personnel in the market can make the adoption journey more difficult both in terms of startup and ongoing operations. Most cloud providers provide consultants for startup, however, the continuance of operations needs in-house personnel for cost effectiveness. Though the number of personnel required to support the operation of the cloud infrastructure is less than that of a traditional data center, the upskilling of existing personnel and hiring new ones can still be challenging. There is still a lack of skilled personnel in the market in both these areas.

IT operational excellence and DevOps factors were primarily identified by interviewees with technical background (3,6–9,12). In addition, many key speeches in AWS Reinvent, Kubecon, and Cloud Native conferences reinforced the concepts. The CPO in a hospitality cloud hosted SaaS provider (3), and an advisory board member of hybrid cloud healthcare SaaS provider (6) outlined the improvements in agility and stability of their services after offloading many operations to cloud providers so that engineering teams could get more focused in implementing business features. This was also mentioned by the Director of design and architecture (7). The person outlined that the improvements achieved in the engineering process by using out-of-the-box SaaS and PaaS solutions offered by the cloud providers. The person elaborated on the automatic scaling and the

automation facilities of the cloud providers towards reducing implementation time and eliminating many operational downtimes of the system. The Technology Evangelist (13) mentioned many improvements realized by the organization and its clients through the use of containers and microservices architecture. Interviews with the hybrid cloud users (8,9,12) focused on the challenges of these implementations and relative benefits using a hybrid approach to HIS implementation. Even though the hybrid cloud solutions owners (8,9) agreed on the benefits, they pointed out that vendor-lock to a cloud provider is a major concern. They also mentioned the usage of many opensource solutions that can be used with these providers to prevent the lock-in. This can definitely lead to complexities in system design in addition to the need of highly skilled and experienced personnel to manage such configurations. The availability of skilled and experienced personnel was the major obstacle that prevented a hospital with locally hosted HIS to migrate to the cloud (12). The CIO and CMO (15,16) also made repeated mention of the same fact. The paucity in availability of personnel is also the experience of one of the authors throughout his career.

The following hypotheses are related to IT operational excellence and DevOps and cloud computing adoption:

**Hypothesis 2 (H2).** *IT Operational excellence and DevOps influence the Cloud computing adoption for HIS.*

**Hypothesis 2-1 (H2-1).** *Offloading IT operations to cloud providers positively influences cloud computing adoption for HIS.*

**Hypothesis 2-2 (H2-2).** *Automatic scaling of cloud services positively influences cloud computing adoption for HIS.*

**Hypothesis 2-3 (H2-3).** *Already built SaaS and PaaS of cloud providers positively influences cloud computing adoption for HIS.*

**Hypothesis 2-4 (H2-4).** *DevOps and automation of cloud services positively influences cloud computing adoption for HIS.*

**Hypothesis 2-5 (H2-5).** *Availability of for opensource alternatives for cloud management positively influences cloud computing adoption for HIS.*

**Hypothesis 2-6 (H2-6).** *Use of microservices and containers facilities of cloud providers positively influences cloud computing adoption for HIS.*

**Hypothesis 2-7 (H2-7).** *Vendor Lock-in negatively influences cloud computing adoption for HIS.*

**Hypothesis 2-8 (H2-8).** *Complexities in design and operation phases of systems negatively influences cloud computing adoption for HIS.*

**Hypothesis 2-9 (H2-9).** *Skillset and experience requirement of personnel negatively influences cloud computing adoption for HIS.*

### 4.3. Security, Governance, and Compliance

Information security is a critical issue in HIS. Security breaches in HIS may expose very sensitive data [58] and put patient privacy at risk [59]. Khaloufi et al. [58] also recommend that that data security should be applied to all stages of the data lifecycle to ensure data security though measures such as encryption at movement and encryption at rest. High security costs can be an organizational barrier for overall cloud adoption for HIS and implementation [60], since the infrastructure resides outside the physical boundaries of

the organization. Physical access to infrastructure is an important aspect of security for healthcare organizations. However, using cloud computing can actually reduce the costs of compliance with regulations like Health Insurance Portability and Accountability Act of 1996 (HIPAA) and Health Information Technology for Economic and Clinical Health Act (HITECH). Many cloud providers being cognisant of the security concerns of hosting sensitive data outside the organization provide services that are HIPAA compliant [61]. Operating a operate fully redundant, always on fault-tolerant and secure datacenters is also a part of the HIPPA requirement. However, compliance with some aspects of HIPPA by the cloud providers does not guarantee the compliance of the overall system, and more work is needed for a practical framework to implement HIPAA-compliant HIS on the cloud. Furthermore, relying on pre-built cloud SaaS or PaaS solutions may lead to vendor lock-in [28], which also couples the HIS security with the security state of the cloud provider. Vendor coupling may be overcome using opensource or in-house developed solutions, which transfers the security responsibility to the owner but can prove to be resource intensive. Qualitative studies can be done to arrive at security considerations for adapting cloud SaaS and PaaS solutions and customized opensource and in-house solutions. Another risk that is related with the cloud vendors is the inheritance of security vulnerabilities from the vendor [62]. Vulnerability in a cloud provider tool affects all tenants. This can be a major challenge due to the lack of control on the infrastructure and the platform by the healthcare organization. Research is needed that can examine the agility of the vendors in mitigating such security risks.

Internal threats and data leakage within the cloud may cause violation of data privacy and is a major risk factor for HIS [63]. Patient and medical data ownership is one of the primary legal issues related to HIS. There is no direct interpretation of data ownership [64] and the limits for organizations on the usage of this information. This raises an issue that requires work to arrive at clearer boundaries that define each party's responsibilities in the shared responsibility models of security proposed by the cloud providers [65]. Shared responsibility principle of cloud providers splits the responsibility of protecting the infrastructure and the data of a system between the cloud provider and consumer [66], and clear boundaries needs to be defined. Marakhimov et al. [67] stated that one of the issues regarding healthcare technology and wearables privacy concerns is the lack of governmental regulation about the ownership of the generated data and boundaries to access the user health records. Compliance requirements and regulations may often pose challenges to the use and implementation of HIS in the cloud [35], or may limit the data processing and analysis [58]. To resolve such issues, collaboration between academic and governmental parties in addition to industry leaders is needed to arrive at a framework for secure implementation of HIS and to provide international standards for implementation and responsibility sharing between parties. It should be stated there though that as new regulations and compliance requirements are put in place, the agility provided by DevOps in the cloud computing environment will enable cloud-based HIS to change faster than the traditional HIS.

Multiple sessions at the AWS Reinvent, Kubecon, and Cloud Native conferences focused on how security controls of the cloud providers will improve the security of their clients. There were many discussions on compliance with HIPAA and HITECH explicitly, because these are complicated and expensive to implement and maintain. A CTO in a security solutions company (10) specified some challenges that they face with cloud providers. The first was the ambiguity of definitions of shared responsibility models of some cloud providers. The speed at which some cloud providers mitigate some zero-day attacks may critically affect the safety of the client environments and may cause sensitive data exposure or loss, which may lead to multiple legislative and regulatory concerns (16). The IT manager (12) and CMO (16) confirmed that some of the governmental legislations related to patient records complicate the implementation of cloud-hosted HIS process. All CNO, CEO, and CIO, and the QA manager (4,5,14,15) along with the advisory board member (6) were laser focused on the HIPAA compliance and the security of patient

records and their quality. The Chief Internal auditor (1) also mentioned the possibility to use blockchain to secure both financial and patient data in the cloud.

The following hypotheses are related to security, governance, and compliance (see Figure 1):

**Hypothesis 3 (H3).** *Security, governance, and compliance influence cloud computing adoption for HIS.*

**Hypothesis 3-1 (H3-1).** *Inheritance of security controls and benefits of cloud providers positively influence cloud computing adoption for HIS.*

**Hypothesis 3-2 (H3-2).** *Ability to maintain HIPAA and HITECH compliance in the cloud positively influence cloud computing adoption for HIS.*

**Hypothesis 3-3 (H3-3).** *Shared responsibility model negatively influences cloud computing adoption for HIS.*

**Hypothesis 3-4 (H3-4).** *Sharing of security vulnerabilities with a vendor negatively influences cloud computing adoption for HIS.*

## 5. Research Model and Issues for Additional Research

Based on the review of the literature, this research proposes the cloud computing adoption model for HIS shown in Figure 3. Items that can measure the drivers and challenges were arrived at during the same process that was used to arrive at the dimensions and their associated drivers and challenges. The items presented here are preliminary and may need to be further refined when a survey is ready to be administered to a wider audience. To keep the length of the data collection to a reasonable length and given that industry professionals receive several surveys for completion, the initial proposal is to have one item measured on the 7-point Likert scale for each of the drivers and the challenges. These items are listed below:

Construct: Financial performance and cost of cloud services:

- Q1-1: Cloud computing services reduce the total cost of operations (TCO) for IT.
- Q1-2: Cloud pricing models provide flexibility in infrastructure deployment and cost savings.
- Q1-3: There may be hidden costs of cloud computing that the organization will need to manage.
- Q1-4: Cloud computing can lead to unmonitored on-demand usage, thus increasing costs.
- Q1-5: Service outages of cloud computing providers can lead to an undesirable financial impact.

Construct, IT operational excellence and DevOps:

- Q2-1: Offloading IT operations and refocusing on business is a primary benefit of cloud adoption.
- Q2-2: Automatic scaling of cloud infrastructure is a priority feature for an organization's infrastructure needs.
- Q2-3: Ready-to-use SaaS and PaaS of cloud providers is a feature that is attractive to the organization.
- Q2-4: DevOps and automation facilities of cloud providers will provide the agility needed by the organization's HIS.
- Q2-5: Containers and microservices architecture are the current or target roadmap for architecting application in the organization.
- Q2-6: Opensource alternatives for cloud management will help the organization avoid vendor lock-in and ease migration issues.

- Q2-7: Possible vendor lock-in of a cloud provider is a source of concern.
- Q2-8: Cloud architecture complexities in design and operation phases are a source of concern.
- Q2-9: Limited availability of skilled and experienced personnel is a source of concern.

Construct: Security, governance, and compliance of cloud providers.

- Q3-1: Inheritance of security controls and benefits of cloud providers will alleviate security concerns in the organization.
- Q3-2: Obtaining and maintaining HIPAA and HITECH compliance can be done more cost-effectively and easily when using a cloud provider.
- Q3-3: Shared responsibility model for security is a source of concern for the organization.
- Q3-4: Shared security vulnerabilities with the cloud vendor are a source of concern for the organization.

Construct: Cloud computing adoption for HIS

- Q4-1: The organization is ready to adopt a cloud-enabled/native HIS.

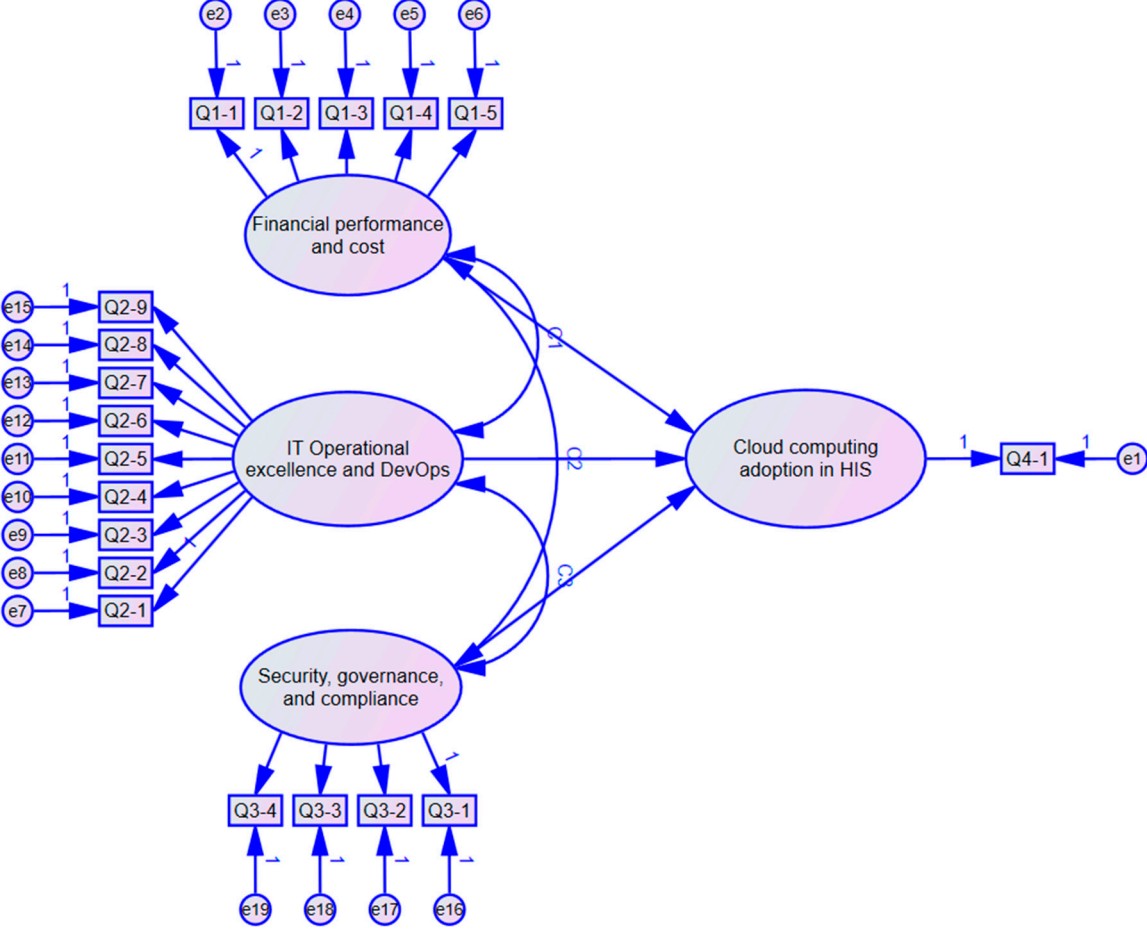

**Figure 3.** Cloud computing adoption model for HIS.

During the process of arrival at the model, some research gaps were identified which have been discussed previously. These gaps pertain more to cloud computing in general but at the same time are also relevant in the context of cloud computing adoption for HIS. These provide an apt opportunity for researchers to further delve into issues that are pertinent to the adoption of cloud computing in HIS and may also apply to a more general case. Studies addressing these issues will aid managers to further make informed decisions

towards the adoption of cloud computing for HIS and in a more general case. These research opportunities are summarized in Table 2. These specific issues and opportunities can be taken up for study by other researchers in the field given that these are based on the experiences of a seasoned professional working in this area.

**Table 2.** Future research directions for dimensions affecting cloud computing in HIS.

| | |
|---|---|
| Financial performance and cost | Assessment of the benefits of using reservation and excess capacity usage pricing schemes over on-demand pricing. Measurement of customer satisfaction on cost and billing of cloud providers. Establishment of models to estimate savings due to the use of cloud providers over the conventional datacenters. |
| IT Operational excellence and DevOps | Quantitative measurement of the performance gains resulting from using cloud infrastructure in comparison to traditional and on-premises data center. Measurement of benefits of hardware generation upgrades. Longitudinal studies comparing scaling provided by well forecasted on-premises hardware to cloud-provided auto scaling features through the use of financial metrics. Measurement of the satisfaction of healthcare institutions of service availability and support of different cloud vendors. Establishment of a framework for HIS implementation using DevOps and microservices with empirical focus on measuring software lifecycle KPIs like delivery time, requirement satisfaction, performance, availability, and scalability, and comparing them to traditional implementations. |
| Security, governance, and compliance | Qualitative research to infer the factors for adapting cloud SaaS and PaaS solutions or opensource solutions in the healthcare context along with associated security risks. Design and implementation of a framework for HIS implementation that complies with international standards with responsibilities definitions. |

## 6. Proposed Model Use for Academic Study

The proposed model has the needed face validity, content validity, and nomological validity on account of the process used to arrive at the model. This process was extensive and has been carried out over a period of about 1 year. However, the empirical validation of the model would be a desired step forward to ground the model in academic rigor. For this purpose, the model and associated hypotheses are planned to be validated using a mixed-methods cross-sectional study that will be conducted on a sample of clinical healthcare organizations in West Bank region in Palestine. Organizations included would be those with an already existing HIS and include a mix of cloud-hosted and self-hosted arrangements. In addition, organizations that are planning to use HIS will also be part of the study. Since these organizations would be evaluating HIS implementations (cloud hosted or self-hosted) on their complete merit, they are likely to provide valuable insights into this study and will benefit from the dimensions and their associated drivers and challenges.

An instrument with the items outlined above has been formulated. This will be pilot tested with a sub sample of the interviewees that were tapped to arrive at the major drivers and challenges. At this stage of development, a single item has been proposed for each of the drivers and challenges. This may be found to be either not clear or capture a concept that is too broad. In such a case, multiple items may be used, and the instrument and model may be refined further. The aim is to keep the instrument short for purposes of easy administration. Following the pilot test, the items will be refined for wording and understandability and then translated to Arabic. Following the translation, another test will be conducted to ensure the understandability of the items. The subjects of these tests will include a panel of experts from healthcare, IT, operations, and financial backgrounds.

The instrument will be administered through a webpage, and electronic communication will be used to reach the participants. An introductory note about the research explaining the purpose and importance of research would be displayed to the candidates on the survey homepage. A pilot test would be conducted on a hospital and a medical complex for a total of 10 employees that will be of different backgrounds and positions.

The primary focus of this pilot test is not on content but more related to the readability, understanding, or navigating of the questionnaire. The pilot would also be helpful in estimation for survey completion time. The needed modifications would be applied to the instrument, and the final form will be used for data collection. The organizations of selected participants will be contacted, and official letters will be delivered introducing the research and explain its importance requesting permission to survey the selected personnel.

Before starting the survey, candidates would be asked for permission to use the data collected from the questionnaire for the purpose of an academic study. To ensure internal validity, a period of 1 month will be selected to collect the data. Major changes in the organizations like change in management, change in key personnel, or HIS changes have very small possibility of occurring during this period. No pretest would be conducted to reduce the learning effect on internal validity. External validity of the results would be ensured as the sample of healthcare organizations have the same operating conditions and environment as other healthcare organizations that are not part of the sample. The pilot candidates would be excluded from the sample to ensure external validity. A follow-up email reminder will be sent after 2 weeks. Structural equation modeling (SEM) with SPSS AMOS will be used to test the significance of the proposed model, and multiple tests like Chi-square and RMSEA will be conducted to determine the confirmation of the hypotheses. Other analyses like reliability of the constructs will also be undertaken.

### 7. Conclusions and Future Research

The research suggests that the health care industry has made improvements toward integrating their HIS locally. However, several HIS issues still need to be addressed. The IT infrastructure and the HIS are there, but optimal utilization of the system for better management of patient care is suspect. There is dissatisfaction with the current patient records systems especially in areas like interface, time taken to input data, and others. Cloud-based HIS may alleviate some of the aforementioned issues. Other benefits may include redesigned financial performance and cost procedures to provide more real time processing.

Users, managers, and employees may be unaware and more skeptical of the value-added potential of the cloud-based HIS due to the privacy and security concerns, and issues related to lack of control. Communication may be the key here, and concerted effort is necessary between DevOps and top management to communicate and educate end-users and to help them become aware of the value-added potential of the cloud-based HIS. Enhanced IT operational excellence can be demonstrated through DevOps by introducing new features frequently. Along with this communication, it is essential to provide additional training to managers, doctors, and employees on the cloud-based HIS to utilize its new features and benefits.

The model presented is based on a rigorous research and has face validity, content validity, and nomological validity. The model can be used by managers and decision makers to guide their adoption decisions for cloud-computing for HIS. Each of the dimensions outlined can be critically evaluated before the adoption decision. Each of the challenges and drivers may themselves be developed into a multiple item scale to allow for more comprehensive evaluation. The HIS area poses some unique challenges due to the sensitive nature of the information stored and its mission's critical nature. This mission's critical nature is an important consideration since it may lead to loss of lives in some cases. The cloud computing model is well proven with service providers like Netflix using it full scale to deliver its services. The booming trend of cloud computing adoption in different industries in recent years is driven by the cost, operation, and security benefits that these services provide. Benefits to the healthcare sector should be no exception but the required regulations and facilitations may need to take place in order to modernize HIS implementation in a reliable and secure fashion. It is time that HIS transitioned to a cloud computing model, and this may be enabled with some further research in the topics outline in this research.

There are other trends and developments that may aid the acceleration of adoption of cloud computing for HIS. Security as mentioned is an important consideration for the EHR repository. Blockchain and smart contracts may provide a solution. Blockchain technology solutions have been used to address several issues, such as security, data privacy, authentication, interoperability, inaccessibility, and stored patient or provider data, in various fields of the healthcare application sector [68]. Several challenges in the use of blockchain though still persist. These include block sizes and block propagation [69], scalability [70], query latency and performance [71], amongst others. Several designs and frameworks have been proposed for blockchain-based EHR [72,73]; the implementations in mainstream HIS used by healthcare organizations have been somewhat limited. Several solutions have been proposed for the challenges posed by blockchain technology [69–71]. In addition, the solutions and frameworks come with stream use of blockchain with and within HIS and will be facilitated by cloud computing given that blockchain by nature as a distributed computing technology is better implemented using cloud technologies.

Another trend likely to drive the adoption of cloud computing for HIS is the use of wearables and Internet of Things (IoT) sensors and devices. IoT sensors and devices, and wearables can be used for remote patient monitoring used in conjunction with video and/or in-person care for a variety of interventions and care outcomes [74]. The wearables and IoT sensors generate data at regular intervals, which needs to be stored and analyzed in real time to flag anomalies that may require immediate attention. This data when aggregated across all patients has all the characteristics of big data as discussed earlier. Big data storage, processing, and analysis invariably require resources, which may be efficiently provided using cloud computing. As such, integration of wearables, and IoT devices and sensors into HIS will be facilitated by the adoption of cloud computing for HIS. Some frameworks for the integration of wearables and IoT devices have been proposed [75], however, as mentioned earlier, the regulatory frameworks for security, privacy, and appropriate use [76] need to be in place before such coupling can happen.

The model presented in this research is proposed to be tested in Palestine by one of the authors, and based on the testing, the model may be refined and tested in US and India. Results are expected to be similar in US and Palestine, but perhaps not for India, which has a good supply of IT personnel and is a significant source of infrastructure management skills. It is also hoped that some other researchers may use this model in other countries and may refine and modify this model based on their own additional insights. Models tested in different countries can be compared and contrasted. It is quite possible that given the economic and political situations in different countries, the results may be quite different and provide other useful insights. Research may be conducted on the issues outlined in Table 2, which can further help the decision makers in cloud adoption decisions.

**Author Contributions:** Conceptualization, A.A.-M.; formal analysis, A.A.-M. and P.C.; methodology, A.A.-M. and P.C.; validation, A.A.-M., P.C. and J.A.R.; writing—original draft, A.A.-M.; writing—review & editing, P.C. and J.A.R. All authors have read and agreed to the published version of the manuscript.

**Funding:** This research received no external funding.

**Institutional Review Board Statement:** Not applicable.

**Informed Consent Statement:** Not applicable.

**Data Availability Statement:** Empirical data collection is planned.

**Conflicts of Interest:** The authors declare no conflict of interest.

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
