# Peer review of "A Model for Examining Challenges and Opportunities in Use of Cloud Computing for Health Information Systems"

_asi, doi:10.3390/asi4010015_

Round 1

Reviewer 1 Report

This article is accepted without any changes. 

Author Response

Dear Reviewer #1:

The introduction has been improved to explain the structure of the papers as per comments from another reviewer. The abstract has also been modified to reflect the goal of this study. 

Details have been added to the research design in outlining the profile of the interviewees who were interviewed for this manuscript. The contribution of each of the interviewee has also been detailed in the manuscript. 

The authors would like to thank the reviewer for the positive review.

Reviewer 2 Report

The authors undertook to contribute to the topic of using the potential of the Cloud computing in the Health Information Systems. The reason for the topic choice is well explained as the Cloud computing can improve some critical aspects of data availability for the HIS, but at the same time there are various issues concerning the security of health data processed in the Cloud, cost of operations and need for experienced personnel to manage the transition from traditional to the Cloud based operations.

Even if the goal is loosely stated it was more or less achieved as the authors provided the model that can be used by various stakeholders to assess the validity (and readiness) of their organization to shift their HIS toward Cloud based solutions. That stated, the study would benefit from the goal being clearly stated in the abstract or in the Introduction part of the paper.

The literature review, being an important part of the research, was performed with due diligence and the appropriate references for the used assumptions are provided.

Paper is structured accordingly and allows the reader to follow the argumentation of the authors, starting with the rationale behind the chosen topic, through the explanation of used methodology and the achieved results, ending with the conclusions and recommendation for further work (especially stating the intent to validate the proposed methodology in various countries). Lack of description of the intended structure of the paper in the introduction may be considered as a minor flaw.

Another, more serious flow is the ambiguous description of the interviews with the experts, which are stated by the authors to be an important part of model development. There is an iteration of who constitutes to be an expert, and how the interviews were conducted but there are no descriptions about how many different experts from stated fields were approached and in which areas their contribution mattered for the purpose of the developed model. Only the contributions of one of the authors that has experience in the field were mentioned on several occasions (like with the issue of lack of skilled and experienced personnel). If the interviews were important for the development of the model, both the sample (i.e., how many experts and from what fields) and what their exact contribution was should be stated clearly in the text. In the current form, the argumentation is mostly drawn upon the literature with minor contributions from one of the authors with the practical experience from the HIS area.

Limitations of the study are clearly described and discussed together with the potential further research directions. The additional value of the reviewed paper is the multiple potential research gaps identified and presented in the paper concerning different aspects of the proposed model. The openness of the authors to share the developed model and their hope for its further usage by different researchers is commendable.

As a novel and original approach to the attempt of providing a scientifically-based tool aimed at assessing the various aspects of implementing the Cloud computing for the HIS the paper is certainly an interesting one for anyone interested in that issue either for scientific or operational use. It would be interesting to see the further results obtained from using the proposed model, either concerning its validation (and possible rework) or planned research in the US and Palestine.

The language could use some improvements, even if it allows the reader to follow the authors’ argument for most parts, there are some minor errors which would require attention from authors and could use language verification, e.g. in line: 54, 89, 94, 218, 242-243, 271, 284, 303-304, 319, 353-354, 410, 432, 434, 442, 448, 469, 485-486, 492-493, 555, 560, 563, 571-572, 589, 608, 624-625, 634-635.

Author Response

The authors undertook to contribute to the topic of using the potential of the Cloud computing in the Health Information Systems. The reason for the topic choice is well explained as the Cloud computing can improve some critical aspects of data availability for the HIS, but at the same time there are various issues concerning the security of health data processed in the Cloud, cost of operations and need for experienced personnel to manage the transition from traditional to the Cloud based operations.

Even if the goal is loosely stated it was more or less achieved as the authors provided the model that can be used by various stakeholders to assess the validity (and readiness) of their organization to shift their HIS toward Cloud based solutions. That stated, the study would benefit from the goal being clearly stated in the abstract or in the Introduction part of the paper.

The abstract has been modified to reflect the goal of this study.

The literature review, being an important part of the research, was performed with due diligence and the appropriate references for the used assumptions are provided.

Paper is structured accordingly and allows the reader to follow the argumentation of the authors, starting with the rationale behind the chosen topic, through the explanation of used methodology and the achieved results, ending with the conclusions and recommendation for further work (especially stating the intent to validate the proposed methodology in various countries). Lack of description of the intended structure of the paper in the introduction may be considered as a minor flaw.

Structure of the paper has been explained in the introduction section.

Another, more serious flow is the ambiguous description of the interviews with the experts, which are stated by the authors to be an important part of model development. There is an iteration of who constitutes to be an expert, and how the interviews were conducted but there are no descriptions about how many different experts from stated fields were approached and in which areas their contribution mattered for the purpose of the developed model. Only the contributions of one of the authors that has experience in the field were mentioned on several occasions (like with the issue of lack of skilled and experienced personnel). If the interviews were important for the development of the model, both the sample (i.e., how many experts and from what fields) and what their exact contribution was should be stated clearly in the text. In the current form, the argumentation is mostly drawn upon the literature with minor contributions from one of the authors with the practical experience from the HIS area.

Comment is very well taken and thank you for your observation.

We have provided in Table 1, the profile of all the experts who we talked for the purposes of this paper. We have also included the names of the conferences where sessions and informal interactions took place. We have also included for each dimension a brief detail of how the interviewees contributed to the elements of the discussion that has been detailed in that section, towards the end of the section before the hypotheses. The discussions with the experts were the starting and primary source of information which we then strengthened with possible academic literature.  We tried to provide as much support from academic literature due to the academic nature of the paper.

Limitations of the study are clearly described and discussed together with the potential further research directions. The additional value of the reviewed paper is the multiple potential research gaps identified and presented in the paper concerning different aspects of the proposed model. The openness of the authors to share the developed model and their hope for its further usage by different researchers is commendable.

As a novel and original approach to the attempt of providing a scientifically-based tool aimed at assessing the various aspects of implementing the Cloud computing for the HIS the paper is certainly an interesting one for anyone interested in that issue either for scientific or operational use. It would be interesting to see the further results obtained from using the proposed model, either concerning its validation (and possible rework) or planned research in the US and Palestine.

The language could use some improvements, even if it allows the reader to follow the authors’ argument for most parts, there are some minor errors which would require attention from authors and could use language verification, e.g. in line: 54, 89, 94, 218, 242-243, 271, 284, 303-304, 319, 353-354, 410, 432, 434, 442, 448, 469, 485-486, 492-493, 555, 560, 563, 571-572, 589, 608, 624-625, 634-635.

We have gone through the paper and checked for English and sentence structure. Hopefully this has alleviated the problem.

Reviewer 3 Report

The manuscript studies the opportunities and challenges of using cloud computing in healthcare information systems. The topic is interesting. But some major issues are as follows:
- "In" is used twice in title, which is not elegant. The second "in" can be replaced by "for"?
- The abstract is not informative. In particular, it only explains the benefits of the research without giving the detailed description. For example, which three dimension? What is the proposed model?
- Capitalizations and abbreviations are used ambiguously. It will make the manuscript hard to read. For example, why "I" in "information technology" is capitalized?
- The authors should summarize the main contributions in introduction.
- Healthcare information exchange should be a very important part of healthcare information system while not mentioned in the manuscript. Some related papers such as "Blochie: a blockchain-based platform for healthcare information exchange".
- The future directions are not well explained. For example, blockchain and Internet of Things can be integrated with cloud computing to provide healthcare information system with better security and reliability. Some example blockchain papers about its concept and privacy preservation are "fairness-based packing of industrial IoT data in permissioned blockchains" and "privacy-preserving and efficient multi-keyword search over encrypted data on blockchain".
- The opportunities and challenges of cloud computing for healthcare information systems as listed in Fig. 1 are more focused on cloud computing while less relevant to "healthcare". The authors should explain more about the usage of cloud computing in healthcare information systems rather than merely about cloud computing.

Author Response

The manuscript studies the opportunities and challenges of using cloud computing in healthcare information systems. The topic is interesting. But some major issues are as follows:
- "In" is used twice in title, which is not elegant. The second "in" can be replaced by "for"?
- The abstract is not informative. In particular, it only explains the benefits of the research without giving the detailed description. For example, which three dimension? What is the proposed model?

The second in has been replaced by for and changes have been made in the text to reflect the same.

The dimensions have been detailed in the abstract and details of the model are provided.

- Capitalizations and abbreviations are used ambiguously. It will make the manuscript hard to read. For example, why "I" in "information technology" is capitalized?

That manuscript has been studied again to alleviate this issue. Wherever possible abbreviations are used in capitalized form.

- The authors should summarize the main contributions in introduction.

This we believe has been done in the problem and significance section.

- Healthcare information exchange should be a very important part of healthcare information system while not mentioned in the manuscript. Some related papers such as "Blochie: a blockchain-based platform for healthcare information exchange".

We have added text on Health Information Exchanges (HIE) in the text on page 3 where the concept of use of computer networking is being discussed.

- The future directions are not well explained. For example, blockchain and Internet of Things can be integrated with cloud computing to provide healthcare information system with better security and reliability. Some example blockchain papers about its concept and privacy preservation are "fairness-based packing of industrial IoT data in permissioned blockchains" and "privacy-preserving and efficient multi-keyword search over encrypted data on blockchain".

We have added future directions with respect to use of Blockchains with HIS and IoT with HIS.

- The opportunities and challenges of cloud computing for healthcare information systems as listed in Fig. 1 are more focused on cloud computing while less relevant to "healthcare". The authors should explain more about the usage of cloud computing in healthcare information systems rather than merely about cloud computing.

Comment is well taken, and thank you for your comment. Since the topic relates to adoption of cloud computing in HIS, the issues outlined though in general relate to cloud computing but at the same time are also applicable to HIS when considering adoption of cloud computing in HIS. Text has been modified to reflect this.

Round 2

Reviewer 3 Report

The authors have well addressed the previous-round comments.